# Comparison of Biocompatibility of 3D-Printed Ceramic and Titanium in Micropig Ankle Hemiarthroplasty

**DOI:** 10.3390/biomedicines12122696

**Published:** 2024-11-26

**Authors:** Si-Wook Lee, Donghyun Lee, Junsik Kim, Sanghyun An, Chul-Hyun Park, Jung-Min Lee, Chang-Jin Yon, Yu-Ran Heo

**Affiliations:** 1Department of Orthopedic Surgery, Dongsan Medical Center, Keimyung University, Daegu 42601, Republic of Korea; shuk@dsmc.or.kr (S.-W.L.);; 2Preclinical Research Center, Daegu-Gyeongbuk Medical Innovation Foundation (K-MEDI hub), Daegu 41061, Republic of Korea; 3Department of Orthopaedic Surgery, College of Medicine, Yeungnam University, Daegu 42415, Republic of Korea; 4Industry-Academic Cooperation Foundation, Keimyung University, Daegu 42601, Republic of Korea; 5Division in Anatomy and Developmental Biology, Department of Oral Biology, Human Identification Research Institute, BK21 FOUR Project, Yonsei University College of Dentistry, 50-1 Yonsei-ro, Seodaemun-gu, Seoul 03722, Republic of Korea

**Keywords:** ankle arthritis, three-dimensional print, artificial joint, total talus replacement surgery

## Abstract

Background: Ankle arthritis is a common degenerative disease that progresses as cartilage damage in the lower tibia and upper talus progresses, resulting in loss of joint function. In addition to typical arthritis, there is also structural bone loss in the talus due to diseases such as talar avascular necrosis. Total talus replacement surgery is the procedure of choice in end-stage ankle arthritis and consists of a tibial, talar component and an insert. However, in cases of severe cartilage and bone damage to the talar bone with less damage to the tibial cartilage, a talar component hemiarthroplasty may be considered. Although the application of total talus replacement surgery using ceramics has been studied, reports on the application of metal 3D printing technology are limited. We aimed to investigate the feasibility of partial talar components using ceramic and titanium 3D printing technology in terms of biocompatibility and stability through animal experiments. Methods: Preoperative 3D CT was acquired and converted to STL files to fabricate a partial talus component for ankle hemiarthroplasty using ceramic and titanium. Six minipigs with an average age of 17 months were implanted with three ceramic (C-group) and three titanium talar components (T-group) in the hind limb ankle joint. The surgery was performed under anesthesia in a sterile operating room and was performed by two experienced foot and ankle specialist orthopedic surgeons. Blood analysis and CT were performed before surgery and every month for 3 months after surgery to assess the extent of inflammatory response and physical stability, sacrifices were performed 3 months after surgery, and H&E staining and micro-CT analysis were performed to compare histological biocompatibility. A grading score was calculated to semi-quantitative assess and compare the two groups. Results: In the postsurgical evaluation, blood analysis revealed that both groups had increased white blood cell counts on the postoperative day after surgery. The white blood cell count increased more in the titanium group (1.85-fold) than in the ceramic group (1.45-fold). After 3 months, all values normalized. During the study, CT analysis confirmed that all artificial samples were displaced from their initial positions. In micro-CT analysis, the adhesive tissue score of the ceramic artificial sample was better than that of the titanium sample (average threshold = 3027.18 ± 405.92). In histologic and grading scores for the inflammatory reactions, the average inflammation indices of the ceramic and titanium groups were 2.0 and 1.21, respectively. Also, the average grade score confirmed based on the results of fibrous tissue proliferation and new blood vessels was 18.4 in the ceramic application group and 12.3 in the titanium application group. Conclusions: In conclusion, both titanium and ceramics have excellent biocompatibility for artificial joints, and ceramic materials can be used as novel artificial joints. Further research on the strength and availability of these ceramics is required.

## 1. Background

In ankle arthritis, conservative treatments often reduce pain and improve joint function. However, when symptoms become severe, surgical intervention, such as total talus replacement surgery, may be necessary [1,2,3,4,5]. Total talus replacement surgery involves the implantation of an artificial joint to replace the damaged talus. While total talus replacement surgery has advantages in terms of reoperation rates, it also presents complications, such as the lack of fusion with existing bone due to the necessity of cutting bone and implanting an artificial joint [6]. In cases of severe cartilage and bone damage to the talar bone with less damage to the tibial cartilage, a talar component hemiarthroplasty may be considered [7,8,9]. Furthermore, 3D-printed arthroplasty, which allows for the replacement of only the damaged part with a custom-made implant, is gaining attention [10,11]. These custom implants are designed to fit precisely, providing stability, preventing degenerative changes in adjacent joints, and appropriately dispersing pressure [8].

Titanium, especially in its pure form (Grade 2), is widely used due to its excellent mechanical strength, high fatigue resistance, and superior biocompatibility. It is particularly favored in load-bearing applications such as joint replacements because of its ability to withstand high stresses and its long-term performance in the human body. However, titanium is susceptible to wear over time, which can lead to the generation of particulate debris, potentially causing adverse biological reactions, such as inflammation and osteolysis. Furthermore, titanium implants can sometimes be prone to corrosion in aggressive biological environments, which may compromise their durability [12].

On the other hand, ceramics, such as alumina and zirconia, offer exceptional hardness, wear resistance, and biocompatibility. These materials have a lower wear rate compared to titanium, making them an ideal choice for patients with higher activity levels or those requiring joint replacement procedures where long-term implant durability is crucial. Ceramics also tend to be less reactive with surrounding tissues, which reduces the risk of inflammation and tissue degradation. However, ceramic materials are more brittle than titanium and may be prone to fracture under high stress or impact, making them less suitable for certain load-bearing applications. Despite these challenges, advancements in ceramic materials, such as improved toughness and the development of composite ceramics, have made them increasingly viable for use in high-performance orthopedic implants. To problem the challenge of maintaining the shape of autogenous and allograft bone, there is an urgent need to develop 3D printing technology that can produce patient-specific implants using bioceramic materials [13,14,15,16,17,18].

This study utilized a 3D-printed ceramic and titanium bone joint model applied to a micropig to establish a bone defect animal model. The aim was to evaluate the safety and effectiveness of a customized artificial ankle joint.

## 2. Materials and Methods

### 2.1. Materials and Equipment

The artificial joint materials used are bioceramic (Ai_2_O_3_) (CGBio, Seoul, Republic of Korea) and titanium (pure Ti grade2) (Institute of Advanced Convergence Techology, Kyung-pook University, Daegu, Republic of Korea). The bioceramic composition, designed for FDM (Fused Deposition Modeling) 3D printing, is a paste-like material consisting of ceramic powders with CaO and SiO_2_ as primary components, combined with a binder in a proportion of 30–50 wt%. This composition is protected under Intellectual Property Registration No. 10-1912839. The paste formulation is specifically optimized for extrusion-based 3D printing and is injected into an FDM 3D printer equipped with a dual-nozzle system. After modeling the CT image, the specimen produced based on 3D printing technology was converted into an STL file using a 3D scanner and extracted through modeling software. The modeling and 3D scan dimensional effectiveness was 83% (CGBio, Seoul, Republic of Korea). The titanium is printed by SLM 280 (SLM solution, Lübeck, Germany).

C-arm (DK Medical Solutions, Seoul, Republic of Korea), an anesthesia machine (Dräger, Germany), and a patient monitoring system (Dräger, Germany) were utilized to perform the surgical procedure. The computed tomography (CT) system (Siemens, Forchheim, Germany) was used to design artificial joints, plan surgery, and monitor stability after implantation.

### 2.2. Three-Dimensionally Printed Artificial Joint Implantation to Talus Defect in Micropigs

Six male micropigs, aged 17 months, with an average weight of 31.17 ± 2.96 kg, were obtained from APURES Co., Ltd. (Pyeongtaek-si, Republic of Korea). The experimental pigs were housed individually in stainless-steel (SUS) cages (1120W × 1620D × 1850H mm) within a dedicated pig housing room on the first floor of the large animal research area. The environmental conditions were maintained at 22 ± 1 °C and 50 ± 10% humidity, with 10–20 air changes per hour. A 12 h light–dark cycle (07:00–19:00) was implemented, and the illumination intensity was kept at 150–300 lux. The pigs were fed “Experimental Pig Feed 1” (Model No. 238075) supplied by Cargill Agri Purina Inc., with the feed quality verified using certificates provided by the supplier. Reverse osmosis (RO) water, purified via a microfiltration system, was provided ad libitum, and the water quality was tested biannually for compliance with potable water standards (58 parameters) by the Daegu Environmental Health Research Institute. The micropigs were divided into two groups: ceramic and titanium application groups (groups C and T, respectively). Group C consisted of micropigs C1–C3, which received 3D-printed ceramic talus joints. Group T consisted of micropigs T1–T3, which received artificial titanium joints (Figure 1). All animal procedures were performed using the protocols approved by the Institutional Animal Care and Use Committee of Daegu-Gyeongbuk Medical Innovation Foundation (Approval number: DGMIF-21052001-01). All procedures were conducted in compliance with the ARRIVE (Animal research: reporting of in vivo experiments) guidelines (https://arriveguidelines.org/, accessed on 1 March 2019), and all efforts were made to minimize the number of animals and induced pain.

The surgery was performed under anesthesia in a sterile operating room and was performed by two experienced foot and ankle specialist orthopedic surgeons (LSW, PCH). Surgery was performed to replace the talus of the left hind limb of each micropig. Prior to surgery, 3D reconstruction of the talus was performed using a CT scan of each micropig. The surgical procedure adhered to the approved protocols and guidelines, ensuring animal welfare and minimizing pain. Anesthesia was induced by an intramuscular injection of Zoletil (tiletamine/zolazepam, 15 mg/kg) and Rompun (5 mg/kg). Anesthesia was maintained with isoflurane at a concentration of 1.5–2%. Hair was removed around the ankle of the left hind limb, followed by disinfection with alcohol and povidone. Ankle incisions were performed anteriorly. The surrounding tissues were carefully incised to expose the talus. A sawing guide (CGBio, Seoul, Republic of Korea) specifically manufactured for the middle of the talus was inserted to facilitate the horizontal cutting of the talus. The proximal half of the talus was removed and an artificial joint was inserted. The artificial joint and the remaining bone were securely fixed by inserting a headless screw through the hole created in the sample. The ligaments, muscles, and skin tissues were sutured, and the surgical site was disinfected with alcohol and povidone. Splints were used to support the joints (Figure 2).

### 2.3. Visual Inspection and Blood Test

The collected blood samples of animals during the experimental period were measured by a hematology blood analyzer (Siemens, Germany) for complete blood count analysis. The body weights of the micropigs were measured before artificial joint application, after 1 day, and at monthly intervals for 3 months. Complete blood tests (CBCs) were performed at the same intervals for 3 months, and white blood cell (WBC), lymphocyte (LYM), and red blood cell (RBC) counts were measured.

### 2.4. Radiographic Analysis

Regular CT scans were performed to observe changes in the condition after artificial joint implantation, and X-rays were used for frequent monitoring of the sample application. Micro-CT imaging (Quantum FX, PerkinElmer, Waltham, MA, USA) was conducted after tissue collection to assess the connection between the artificial joint and the surrounding tissue. Micro-CT parameters were set at 90 kVp, 180 µA, FOV 24 mm, and voxel size 46 µm. The analysis criterion was the ratio (%) of the remaining tissue area to the contact surface of the implanted bone tissue (Figure 3).

### 2.5. Histological Analysis

After removing the artificial joint samples, the surrounding tissues were fixed in 10% neutral buffered formalin (Epredia™, Thermo Fisher Scientific, Waltham, MA, USA) for 2 weeks. The fixed specimens were decalcified using a 0.5 M EDTA solution at 40 °C for 8 weeks. Decalcified specimens were embedded in paraffin and prepared as tissue slides. Tissue slides were stained with hematoxylin and eosin (H&E) to evaluate local biological effects. The degree of inflammation in the joint area was classified based on the infiltration of inflammatory cells and categorized as minimal (1+), mild (2+), moderate (3+), or severe (4+). Infiltration of inflammatory cells and tissue reactions, such as generation of new blood vessels, fibrosis, and fat invasion, were evaluated based on the ISO10993-6 guidelines and classified as minimal infiltration (1+), mild (2+), moderate (3+), or heavy infiltration (4+). The presence of the transplanted material was confirmed, and the degree and characteristics of inflammation related to the transplanted material were described.

### 2.6. Statistical Analysis

For the in vivo stability test, three artificial joint samples were applied on the micropig talus defect model per group. In addition, statistical analysis was performed using a two-way analysis of variance (ANOVA) with Turkey’s multiple comparison post hoc tests. All values are expressed as mean ± standard deviation and all the experimental groups are compared with each other.

## 3. Results

### 3.1. Weight Measurement and Visual Inspection

Before the artificial sample application, the average body weight was 33.00 ± 1.00 kg in group C and 35.67 ± 2.31 kg in group T. One month after application, the body weight in group C remained similar to that before application (33.67 ± 1.15 kg), whereas group T showed a weight reduction of approximately 2 kg (32.67 ± 1.15 kg). However, according to an independent t-test, they were not significantly different (*p* = 0.44). After 2 months of application, both groups C and T showed an increase in weight compared to that before application, with average weights of 36.33 ± 0.58 kg and 36.67 ± 4.73 kg, respectively. Initially, all subjects experienced difficulty in walking and feeding. Nutrients were administered based on individual conditions. After 1 month, signs of recovery were observed in all subjects, with normal healing of the incision site and no swelling or inflammation. However, the subjects avoided placing weight on the leg that had undergone surgery and did not walk normally. After 2 months, it was confirmed that the legs with the artificial samples temporarily stood on the floor. However, normal walking using the leg with the artificial sample remained impossible (Figure 4).

### 3.2. Blood Test

Blood samples were collected at various time points: before artificial sample application and 1 day, 1 month, 2 months, and 3 months after artificial sample application. On day 1 of sample application, both groups C and T showed a significant increase in the white blood cell (WBC) count compared to that before application. In group C, the WBC increased from 8.18 (×10^3^ cells/µL) to 11.91 (×10^3^ cells/µL). In group T, it increased from 8.98 (×10^3^ cells/µL) to 16.64 (×10^3^ cells/µL). By month 3, the WBC count in both groups had returned to normal levels, with 7.42 (×10^3^ cells/µL) in group C and 8.82 (×10^3^ cells/µL) in group T (Figure 5).

### 3.3. Autopsy and Radiographic Analysis

CT analysis was performed three weeks after the implant application, revealing that in both group C and group T, the implant had migrated from its initial position. After three months, autopsy of the sacrificed subjects confirmed damage to the thin section of the implant and dislodgement of the securing screw (Figure 6). None of the screws used to fix the samples were attached to the bone tissue. In group C, some fibrous tissue was attached to the sample surface, but most of the micropig was separated from the surrounding tissue and partially damaged. Discoloration was observed in group C. In group T, no tissue was attached to the sample surface, and relatively thick fibrous connective tissue was formed (Figure 7). And bone erosion was found. Observing the joint surface of the artificial sample and talus, adhesive tissue was only confirmed in group C, with measurements of 18.26%, 2.89%, and 0.84% in C1, C2, and C3, respectively (average threshold = 3027.18 ± 405.92) (Figure 8). The bony erosion of the artificial sample is shown in Figure 9. The H&E-stained images demonstrate the presence or absence of residual implanted material in the tissue.

### 3.4. Histological Results

In all subjects, inflammation in the joint area varied among the subjects and ranged from 0 to 4+ (Table 1 and Table 2). The average degree of inflammation was 2.0 in group C and was 1.21 in group T, and it mainly appeared around the necrotic tissue mass, accompanied by the infiltration of lymphocytes, macrophages, and multinucleated giant cells. Tissue reactions included the proliferation of connective tissue and new blood vessels; however, no graft materials were observed in the inflammatory areas. The results of H&E staining are shown in Figure 10. There was no significant difference between the two graft material groups in terms of tissue reactions, such as fibrous tissue proliferation and neovascularization.

## 4. Discussion

In this study, we compared and evaluated the biocompatibility of customized ceramic and titanium artificial ankle joints manufactured using 3D printing technology in micropigs. Our findings indicate that the biocompatibility of ceramic and titanium artificial ankle joints is comparable, with each material presenting unique advantages and challenges.

As the global population ages, enhancing the quality of life through advanced medical treatments becomes increasingly critical. Osteoarthritis and bone defects are significant health issues, driving the demand for innovative biomedical devices [19,20,21,22]. Previous research has established the biocompatibility of ceramic materials for fractures and implants [21,22]. It has shown that the biocompatibility of ceramic and titanium is similar [23,24,25]. Our research results were similar, the results showed no significant differences in biocompatibility between titanium and ceramic.

During the study, we encountered several postoperative management challenges, particularly in maintaining stable feed intake among the micropigs. Notably, the group that received titanium joints (group T) experienced weight loss, possibly due to surgical stress and discomfort [26]. However, there was no significant difference as a result of an independent t-test (*p* = 0.44). The study results showed a significant increase in WBC count on day 1 of sample application in all groups. It is believed that this is due to a rapid increase in inflammation due to the influence of the implant material immediately after surgery [27].

In the previous study, the ultimate shearing force for the ceramic materials was lower than for the implants coated with pure titanium implants [28]. Similarly, in group C, the strength of the sample in the body was relatively low and there was damage due to the screw, but the stress on the remaining talus was reduced and the remaining tissue was well maintained. We were limited to using titanium screws with different physical properties than ceramic to fix the ceramic implants. On the other hand, the titanium application group did not break due to the high strength of the sample, but the external force of the screw was transmitted as is, causing bone tissue resorption [29]. Accordingly, ceramics are vulnerable to physical shock and may fracture or produce fine fragments, so there is a need to check the surrounding tissue through biopsy.

In micro-CT, considering that adherent tissue was observed only in the group C, it is believed that ceramic is advantageous for bonding to surrounding tissues. A recent study comparing titanium with the new plasma electrolytic oxidation (PEO) modified titanium, i.e., similar to ceramic, found that titanium with ceramic-like pores had a higher adhesion rate to surrounding tissue [30]. Ankle hemiarthroplasty in humans targets uncommon disease groups such as talar avascular necrosis. Pre-made off-the-shelf implants are difficult to make due to individual differences, requiring personalized implants. Three-dimensionally printed implants are an alternative, and in younger populations with higher activity levels, we expect ceramic materials with the same biocompatibility as metals to be more suitable, as they can reduce the probability of insert wear [31,32].

In the grade score related to inflammation, no inflammatory cell infiltration or tissue reaction was observed around the remaining graft material, and no graft material was observed in the inflamed area, so it is judged to be a reaction due to tissue damage that occurred during surgery rather than inflammation related to the graft material.

This study had several limitations. First, although this study was conducted on animals, it was conducted by orthopedic surgeons rather than veterinarians. Therefore, we do not know much about the anatomy of the micropig. Therefore, the anatomical structure of the micropig talus posed challenges for stable fixation, and the surgical methods used may need modification. Additionally, the small sample size limits the generalizability of our findings. Future research should focus on improving surgical techniques and increasing the sample size to validate and extend our results.

In conclusion, both ceramic and titanium implants have excellent biocompatibility but have clear pros and cons in terms of strength. In order to utilize ceramics, additional research is needed on appropriate methods, such as conducting biopsies.

## Figures and Tables

**Figure 1 biomedicines-12-02696-f001:**
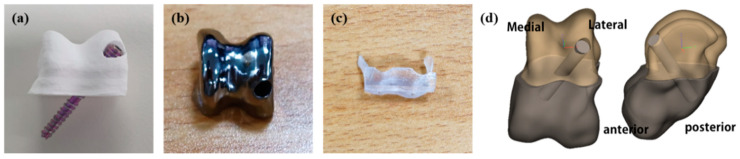
Three-dimensionally printed artificial joint ((**a**): ceramic, (**b**): titanium) and surgical guide (**c**) images. Expected 3D images after jointing bone tissue and artificial joint (**d**).

**Figure 2 biomedicines-12-02696-f002:**
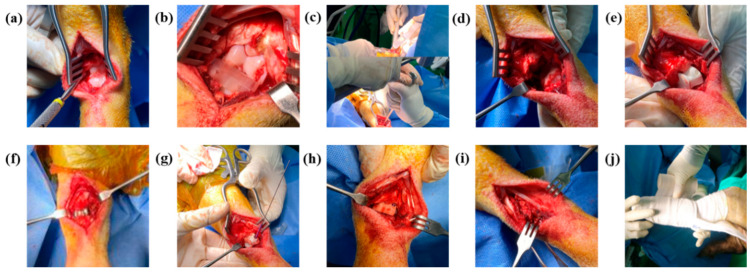
Micropig talus removal and artificial ankle joint application, (**a**) incision, (**b**) surgical guide installation, (**c**) talus cutting, (**d**) talus removal, (**e**) ceramic sample application, (**f**) titanium sample application, (**g**) screw placement, (**h**) sample application, (**i**) suture, (**j**) splint treatment.

**Figure 3 biomedicines-12-02696-f003:**
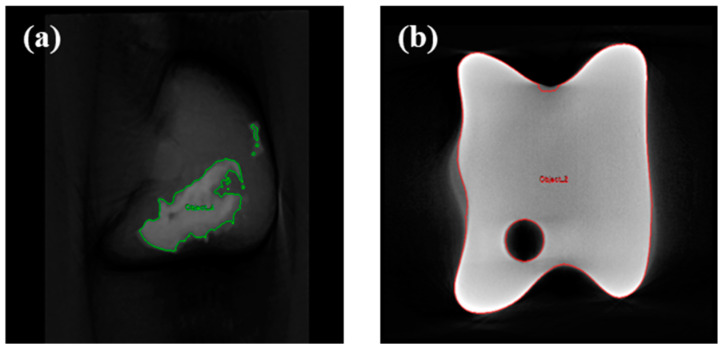
Micro-CT analysis ((**a**): adherent tissue area, (**b**): total area).

**Figure 4 biomedicines-12-02696-f004:**
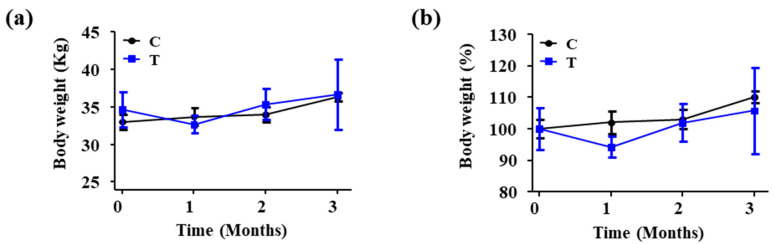
Weight change graph of the subject, (**a**) weight change, (**b**) weight change rate.

**Figure 5 biomedicines-12-02696-f005:**
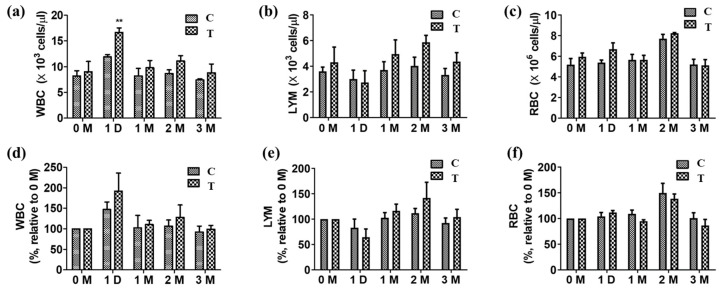
General blood test. (**a**) White blood cell (WBC) concentration (×10^3^ cells/µL), (**b**) lymphocyte (LYM) concentration (×10^3^ cells/µL), (**c**) red blood cell (RBC) concentration (×10^3^ cells/µL) according to the number of months after surgery, (**d**) WBC increase rate, (**e**) LYM increase rate, (**f**) RBC increase rate compared to immediately after surgery (0 months), **: *p* < 0.01 significant differences as compared with C group.

**Figure 6 biomedicines-12-02696-f006:**
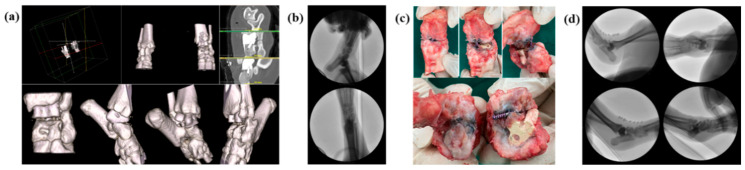
Visual evaluation results, (**a**) CT 3-dimensional reconstruction on day 1 of application, (**b**) C-arm image immediately after application, on week 3 of application, (**c**) autopsy picture and (**d**) C-arm images on week 3 of application.

**Figure 7 biomedicines-12-02696-f007:**
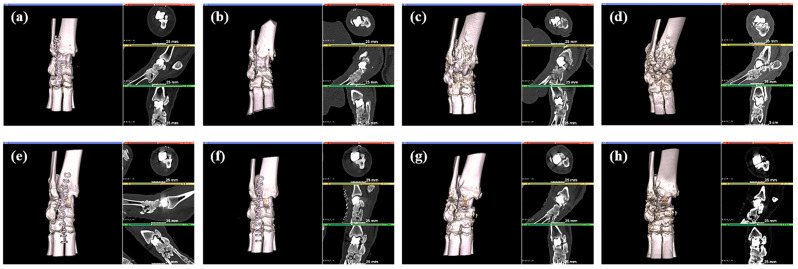
CT images, 1 day (**a**), 1 month (**b**), 2 month (**c**), 3 month (**d**) after ceramic insertion, and 1 day (**e**), 1 month (**f**), 2 month (**g**), 3 month (**h**) after titanium insertion.

**Figure 8 biomedicines-12-02696-f008:**
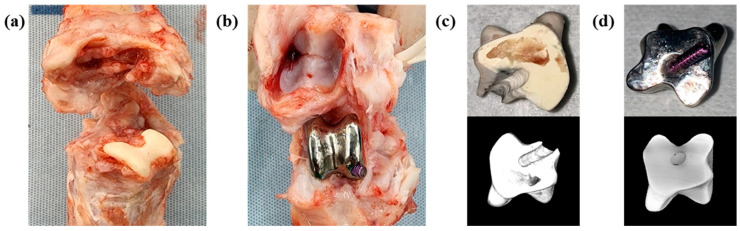
Representative photographic images, (**a**) ceramic applied, (**b**) titanium applied, (**c**) ceramic artificial sample surface image and micro-CT reconstruction, (**d**) titanium artificial sample surface image and micro-CT reconstruction images.

**Figure 9 biomedicines-12-02696-f009:**
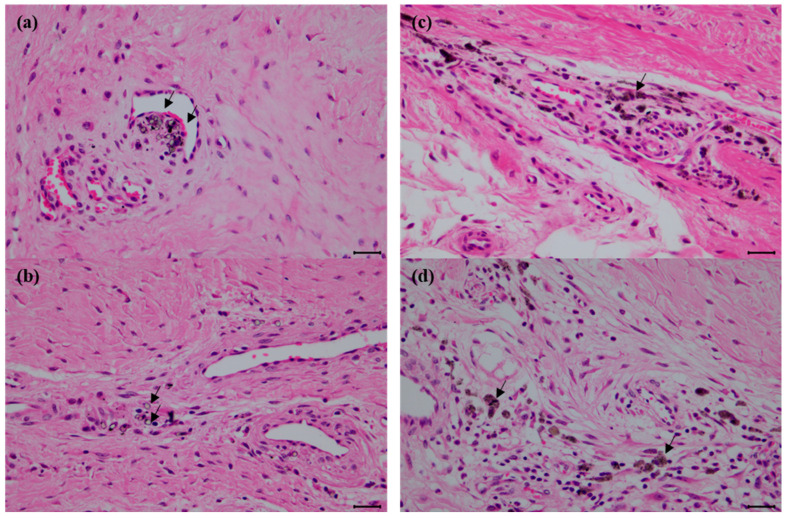
Histological features of the remnant implanted materials in the tissue. Note the scattered clumps of implanted materials (arrow), (**a**,**b**) is ceramics and (**c**,**d**) is titanium. Bars = 25 µm for all.

**Figure 10 biomedicines-12-02696-f010:**
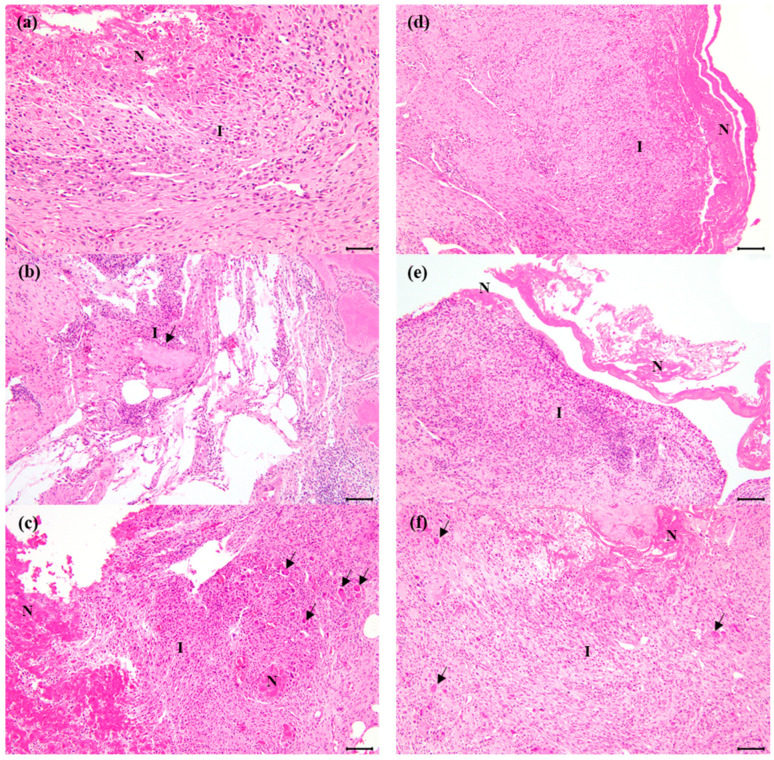
Histological features of the representative joints after 3 months of ceramics and titanium implantation. (**a**–**c**) The ceramic group. (**d**–**f**) The titanium group. Note the necrotic tissue masses (N), surrounded by inflammatory cell and tissue reactions (I). Lymphocytes, macrophages, and multinucleated giant cells (arrows) were infiltrated. H&E bars = 100 μm for except (**a**) and 50 µm for (**a**).

**Table 1 biomedicines-12-02696-t001:** Grading scores of the parameters associated with inflammatory cell infiltration and tissue responses in the joints with ceramics implantation.

Group												
Animal ID	C1-1	C1-2	C1-3	C1-4	C2-1	C2-2	C2-3	C2-4	C3-1	C3-2	C3-3	C3-4
Presence of the test material	P	P	P	P	N	P	N	N	P	P	P	P
Inflammatory cell infiltration, overall *	0	0	0	2+	2+	2+	3+	3+	3+	3+	4+	2+
Mean ± SD	2.0 ± 1.35											
Cell type/response **												
PMN cells	0	0	0	0	0	0	0	0	0	0	0	0
Lymphocytes	0	0	1	1	2	1	3	3	1	1	2	1
Plasma cells	0	0	0	0	2	3	3	3	0	0	1	0
Macrophages	1	1	1	3	2	1	2	2	4	4	4	3
Giant cells	0	0	0	1	1	1	0	0	2	3	1	0
Necrosis	0	0	1	2	2	2	1	1	2	4	3	2
Sub-total (*2)	2	2	6	14	18	16	18	18	18	24	22	12
Mean ± SD	14.2 ± 7.31											
Response **												
Neovascularization	0	0	2	3	2	2	3	3	3	2	3	2
Fibrosis	1	1	3	3	2	2	2	2	2	3	3	2
Fatty infiltration	0	0	0	0	0	0	0	0	0	0	0	0
Sub-total	1	1	5	6	4	4	5	5	5	5	6	4
Mean ± SD	4.25 ± 1.66											
Total	3	3	11	20	22	20	23	23	23	29	28	16
Mean ± SD	18.4 ± 8.64											

* Criteria for the lesion:1, minimal; 2, mild; 3, moderate; 4, severe. **, The evaluation criteria of the cell type/response and response were modified from the evaluation system of ISO 10993-6.

**Table 2 biomedicines-12-02696-t002:** Grading scores of the parameters associated with inflammatory cell infiltration and tissue responses in the joints with titanium implantation.

Group												
Animal ID	T1-1	T1-2	T1-3	T1-4	T2-1	T2-2	T2-3	T2-4	T3-1	T3-2	T3-3	T3-4
Presence of the test material	P	P	N	N	N	N	N	P	P	N	N	N
Inflammatory cell infiltration, overall *	0	3+	2+	2+	0.5	0	0	2+	4+	1+	0	0
Mean ± SD	1.21 ± 1.37											
Cell type/response **												
PMN cells	0	0	0	0	0	0	0	0	0	0	0	0
Lymphocytes	0	1	1	1	1	0	0	2	1	1	0	0
Plasma cells	0	1	1	1	0.5	0	0	0	0	0	0	0
Macrophages	0	3	3	2	1	1	0	2	3	2	0	0
Giant cells	0	1	0	0	0	0	0	0	2	1	0	0
Necrosis	3	3	2	2	0	0	0	1	3	1	1	1
Sub-total (*2)	6	18	14	12	5	2	0	10	18	10	2	2
Mean ± SD	8.25 ± 6.36											
Response **												
Neocascularization	0	3	3	2	1	4	2	2	4	3	2	0
Fibrosis	1	2	2	2	1	4	1	3	4	2	1	0
Fatty infiltration	0	0	0	0	0	0	0	0	0	0	0	0
Sub-total	1	5	5	4	2	8	3	5	8	5	3	0
Mean ± SD	4.08 ± 2.47											
Total	7	23	19	16	7	10	3	15	26	15	5	2
Mean ± SD	12.3 ± 7.88											

* Criteria for the lesion: 0.5, very slight; 1, minimal; 2, mild; 3, moderate; and 4, severe. **, The evaluation criteria of the cell type/response and response were modified from the evaluation system of ISO 10993-6.

## Data Availability

Data are contained within the article.

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
