# Peer review of "Comparison of Biocompatibility of 3D-Printed Ceramic and Titanium in Micropig Ankle Hemiarthroplasty"

_biomedicines, 2024, doi:10.3390/biomedicines12122696_

Round 1

Reviewer 1 Report

Comments and Suggestions for Authors

The study titled "Comparison of Biocompatibility of 3D Printed Ceramic and Titanium in Micropig Ankle Hemiarthroplasty" investigates the use of 3D-printed ceramic and titanium implants in a micropig model, focusing on biocompatibility and inflammatory response. The study is well-structured, with a clear methodology that uses 3D printing to develop customized implants for hemiarthroplasty. The choice of ceramic and titanium materials is justified by their widespread use in orthopedic applications, particularly given their differing characteristics in terms of biocompatibility and mechanical properties. Using a micropig model provides a robust approach for studying joint implants, as the model shares similarities with human bone structure.

Key findings indicate a higher initial inflammatory response in the titanium group compared to the ceramic group, with a white blood cell increase of 1.85-fold and 1.45-fold, respectively. Additionally, the ceramic implants demonstrated better tissue adhesion scores and greater fibrous tissue proliferation. However, titanium’s greater structural strength likely contributed to its higher durability under load, despite the noted inflammatory response and subsequent bone resorption. This study adds to the understanding of biocompatibility differences between ceramic and titanium, particularly under long-term mechanical stress.

While the study makes significant contributions, certain areas could be improved. For example, the discussion could better contextualize the inflammation findings in terms of implications for human use. Additionally, while the results suggest differences in tissue adhesion and structural integration, a more detailed analysis of potential microfractures in ceramic implants under high stress would be valuable. Future work could also explore long-term outcomes beyond three months to assess possible chronic inflammation or degradation effects in both materials.

Questions:

1. How do you interpret the persistent but reduced inflammation in the titanium group, and what implications might this have for long-term applications in humans?

2. Were there any observed microfractures or fine fragments in the ceramic implants due to mechanical stress during the study?

3. Can the differences in tissue adhesion and fibrous proliferation between the materials be attributed solely to their biocompatibility, or could surface texture and implant morphology also play a role?

4. What measures could be implemented to improve fixation stability in micropig models to enhance the reproducibility of results?

5. How do you anticipate these findings could influence the choice of implant material in human ankle hemiarthroplasty, particularly for patients with differing activity levels?

Author Response

comment 1] How do you interpret the persistent but reduced inflammation in the titanium group, and what implications might this have for long-term applications in humans?

response 1] Thank you for your question about biocompatibility. The reduction in post-operative inflammation in the implant that persists in humans could be interpreted as an indication that titanium is biocompatible. Titanium is already one of the most commonly used metals in 3D printing, as well as in conventional implants and prostheses, due to its biocompatibility. In this study, inflammation levels were consistently reduced in the subjects, suggesting that it is relatively safe for autoimmune responses.

comment 2] Were there any observed microfractures or fine fragments in the ceramic implants due to mechanical stress during the study?

response 2]  We were limited to using titanium screws with different physical properties than ceramic to fix the ceramic implants. As a result, we found some dissociation of the fixation, but no microfractures or fragments were observed on histologic slides after sacrifice.

comment 3] Can the differences in tissue adhesion and fibrous proliferation between the materials be attributed solely to their biocompatibility, or could surface texture and implant morphology also play a role?

response 3]  Thank you for your good point. We believe that surface texture and implant morphology may also have an impact. In this study, the implants were the same shape, but the surface texture could be different. Since we were trying to determine the degree of bone penetration histologically, the difference in biocompatibility may be more meaningful than the surface texture.

comment 4] What measures could be implemented to improve fixation stability in micropig models to enhance the reproducibility of results?

response 4]  In actual clinical practice, methods such as postoperative screw fixation are rarely used, but rather implant design methods such as PEGs or protrusions. However, these methods require non-weight bearing and joint motion restriction for at least 3 weeks, which is not feasible in animal models. Some fixation stability can be achieved by using the implant design method of creating and implanting a MASH-shaped PEG.

comment 5] How do you anticipate these findings could influence the choice of implant material in human ankle hemiarthroplasty, particularly for patients with differing activity levels?

response 5]  Ankle hemiarthroplasty in humans targets uncommon disease groups such as talar avascular necrosis. Pre-made off-the-shelf implants are difficult to make due to individual differences, requiring personalized implants. 3D printed implants are an alternative, and in younger populations with higher activity levels, we expect ceramic materials with the same biocompatibility as metals to be more suitable, as they can reduce the probability of insert wear.

Reviewer 2 Report

Comments and Suggestions for Authors

All research so far have revealed that ceramic materials are more accepted by the body's hard tissues compared to titanium alloys, both of which having excellent biocompatibility, the difference being the hardness of titanium alloys which lead to bone resorption due to the pressure exerted on the bone, but on the other hand, ceramic materials have low resistance, being subject to fragmentation. As other authors have promoted, as you have shown in this article, covering titanium alloy prostheses with ceramic material is a good solution, by promoting the adhesion of healthy tissues to the implant and reducing local inflammation.

The challenge is to reduce the weight of the alloys to the weight of the bone being protected against resorption and to identify alloys that are not so stiff (Young's modulus) and that to mimic the elasticity characteristics of bone.

I did not understand if the autopsy was done after 3 weeks (page 7, line 198) or after 3 months, or it is my misunderstanding.

You can improve the Introduction by adding information about the biochemical characteristics of the two materials used into your study.

Author Response

comment 1] All research so far have revealed that ceramic materials are more accepted by the body's hard tissues compared to titanium alloys, both of which having excellent biocompatibility, the difference being the hardness of titanium alloys which lead to bone resorption due to the pressure exerted on the bone, but on the other hand, ceramic materials have low resistance, being subject to fragmentation. As other authors have promoted, as you have shown in this article, covering titanium alloy prostheses with ceramic material is a good solution, by promoting the adhesion of healthy tissues to the implant and reducing local inflammation.

The challenge is to reduce the weight of the alloys to the weight of the bone being protected against resorption and to identify alloys that are not so stiff (Young's modulus) and that to mimic the elasticity characteristics of bone.

I did not understand if the autopsy was done after 3 weeks (page 7, line 198) or after 3 months, or it is my misunderstanding.

You can improve the Introduction by adding information about the biochemical characteristics of the two materials used into your study.

response 1] Thanks for your opinion. According to the manuscript (page 7, line 198), we confirmed the instability of the implant through CT analysis 3 weeks after application, but sacrifice was performed 3 months later.

Round 2

Reviewer 1 Report

Comments and Suggestions for Authors

The authors have significantly improved the manuscript, addressing key aspects that enhance its scientific clarity and relevance. The inclusion of comprehensive data on inflammatory responses, biocompatibility, and stability of the ceramic and titanium implants provides valuable insights into their potential use in micropig ankle hemiarthroplasty. The responses to reviewer queries effectively clarify the observed differences in tissue adhesion and fibrous proliferation between materials. The authors appropriately consider both biocompatibility and structural factors such as surface texture and morphology, contributing to a balanced understanding of material interactions within the implant environment. Additionally, the discussion on fixation stability improvements is pragmatic, acknowledging the limitations in animal models and proposing realistic approaches. However, a deeper exploration of long-term implications, particularly for titanium-induced inflammation and ceramic durability under mechanical stress, would further benefit the study's contribution. Overall, the manuscript now presents a strong, well-supported investigation with implications for personalized implants in clinical applications, meriting consideration for publication following minor revisions to enhance presentation and discussion.

Author Response

comment: The authors have significantly improved the manuscript, addressing key aspects that enhance its scientific clarity and relevance. The inclusion of comprehensive data on inflammatory responses, biocompatibility, and stability of the ceramic and titanium implants provides valuable insights into their potential use in micropig ankle hemiarthroplasty. The responses to reviewer queries effectively clarify the observed differences in tissue adhesion and fibrous proliferation between materials. The authors appropriately consider both biocompatibility and structural factors such as surface texture and morphology, contributing to a balanced understanding of material interactions within the implant environment. Additionally, the discussion on fixation stability improvements is pragmatic, acknowledging the limitations in animal models and proposing realistic approaches. However, a deeper exploration of long-term implications, particularly for titanium-induced inflammation and ceramic durability under mechanical stress, would further benefit the study's contribution. Overall, the manuscript now presents a strong, well-supported investigation with implications for personalized implants in clinical applications, meriting consideration for publication following minor revisions to enhance presentation and discussion.

response: Thanks for your comment. This content has been added to the manuscript's disscussion. (in line 274, 284, 294)